# Optical model for the Baltic Sea with an explicit CDOM state variable: a case study with Model ERGOM (version 1.2)

Thomas Neumann[1], Sampsa Koponen[2], Jenni Attila[2], Carsten Brockmann[3], Kari Kallio[2], Mikko Kervinen[2], Constant Mazeran[4], Dagmar Müller[3], Petra Philipson[5], Susanne Thulin[5], Sakari Väkevä[2], and Pasi Ylöstalo[2]

[1]Leibniz Institute for Baltic Sea Research Warnemünde, Seestr. 15, 18119 Rostock, Germany
[2]The Finnish Environment Institute, Latokartanonkaari 11, 00790 Helsinki, Finland
[3]Brockmann Consult GmbH, Max-Planck-Str. 2, 21502 Geesthacht, Germany
[4]SOLVO, 3 rue Saint-Antoine, 06600 Antibes, France
[5]Brockmann Geomatics Sweden AB, Torshamnsgatan 39, SE-164 40 Kista, Sweden

**Correspondence:** Thomas Neumann (thomas.neumann@io-warnemuende.de)

**Abstract.** Colored dissolved organic matter (CDOM) in marine environments impacts primary production due to its absorption effect on the photosynthetically active radiation. In coastal seas, CDOM originates from terrestrial sources predominantly and causes spatial and temporal changing patterns of light absorption which should be considered in marine biogeochemical models. We propose a model approach in which Earth Observation (EO) products are used to define boundary conditions of CDOM concentrations in an ecosystem model of the Baltic Sea. CDOM concentrations in riverine water derived from EO products serve as forcing for the ecosystem model. For this reason, we introduced an explicit CDOM state variable in the model.

We show that the light absorption by CDOM in the model can be improved considerably in comparison to approaches where CDOM is estimated from salinity. The model performance increases especially with respect to spatial CDOM patterns due to the consideration of single river properties. A prerequisite is high quality CDOM data with sufficiently high spatial resolution which can be provided by the new generation of ESA satellite sensor systems (Sentinel 2 MSI and Sentinel 3 OLCI). Such data are essential, especially when local differences in riverine CDOM concentrations exist.

## 1 Introduction

Colored dissolved organic matter (CDOM) is a major light absorption constituent in the marine environment and especially in coastal seas. The spectral absorption characteristic of CDOM follows an exponential function with highest absorption towards shorter wavelengths (e.g. Nelson and Siegel, 2002). By modifying the underwater light climate, CDOM has an impact on primary productivity, e.g. in clear water sufficient light intensity to enable phytoplankton growth is available down to greater depths than in turbid waters (Dutkiewicz et al., 2015). Water temperature is affected by CDOM absorption as well. In turbid

water, the short wave light absorption is located in the upper water column increasing the temperature while in clear water a thicker layer is warmed but to a lesser degree (Jolliff and Smith, 2014). This process impacts especially the sea surface temperature (SST). Model studies show an SST increase up to 2 K in coastal regions when colored organic materials are considered (Gnanadesikan et al., 2019).

Jerlov (1976) developed a classification for different water masses based on specific optical properties. This classification is widely used in numerical ocean models (e.g., Griffies, 2004). For global models, this parametrization works reasonably well. However, coastal ecosystems with substantial terrestrial runoff require more detailed parametrization of light penetration. Especially variable light attenuation in river plumes and their environs affect the hydrodynamic and ecological response (Cahill et al., 2008).

Marine CDOM comprises humid substances (yellow substances) of terrestrial origin, and autochthonously produced CDOM. Its degradation is governed by photochemical bleaching and bacterial activity. In freshwater dominated systems, like the Baltic Sea, terrestrial CDOM dominates (e.g., Stedmon et al., 2010, and references herein). In such systems, salinity and light absorption due to CDOM show a robust relationship (Neumann et al., 2015). The relationship is hyperbolic indicating the degradation processes.

For coupled physical-biogeochemical models of freshwater influenced coastal seas, a spatially resolved CDOM concentration is important for realistic light climate estimates. Based on the nearly conservative character of CDOM, statistical models have been developed which estimate absorption from salinity and e.g. chlorophyll (Kowalczuk et al., 2006; Neumann et al., 2015). These models usually deliver reasonable results. However, two distinct disadvantages are prominent: (i) uncertainties in model salinity propagate into the biogeochemical model and (ii) variability in CDOM riverine load (Skoog et al., 2011; Asmala et al., 2013) cannot be resolved. These disadvantages can be eliminated by introducing an independent CDOM state variable into the biogeochemical model (Dutkiewicz et al., 2015). A necessary prerequisite are boundary data for riverine CDOM loads of sufficiently good quality which is not commonly available for most rivers (Pefanis et al., 2020).

In this study, we present the implementation of a CDOM state variable in the biogeocemical model ERGOM (Ecological ReGional Ocean Model, Leibniz Institute for Baltic Sea Research (2015)), the generation of CDOM boundary data with the aid of satellite imagery, and we discuss the effect of the proposed model extension on the Baltic Sea ecosystem.

## 2 Methods and data

### 2.1 Model development and description

We start presenting the optical model including an explicit CDOM state variable, its implementations as part of a biogechemical model, and we briefly introduce the circulation and biogechemical model used for this study.

### 2.1.1 Fundamentals of the optical model

Starting point of the development is the optical model as in the study by Neumann et al. (2015). The photosynthetical active radiation (PAR) follows an exponential decay with depth $z$:

$$PAR(z) = PAR(0) \cdot \exp(-K_{PAR} \cdot z), \tag{1}$$

$K_{PAR}$ is the underwater bulk light attenuation and is described by 5 components:

$$K_{PAR} = k_w + k_c \cdot Chl + k_{det} \cdot DET + k_{don} \cdot DON + K_{CDOM}(S), \tag{2}$$

$k_c$, $k_{det}$, and $k_{don}$ are material specific constants and $Chl$, $DET$, and $DON$ are concentrations of chlorophyll, detritus, and dissolved organic nitrogen, respectively. These concentrations are state variables of the ecosystem model ERGOM or, in the case of chlorophyll, can be estimated from model phytoplankton (Sect. 2.1.2). $k_w$ is the attenuation coefficient for pure water. In Neumann et al. (2015), $K_{CDOM}(S)$ is a statistical relationship between *in situ* salinity and CDOM absorption for the Baltic Sea derived from observations (Eq. A. For the new approach, we use the additional state variable CDOM:

$$K_{CDOM} = k_{cdom} \cdot CDOM \tag{3}$$

The PAR attenuation now reads:

$$K_{PAR} = k_w + k_c \cdot Chl + k_{det} \cdot DET + k_{don} \cdot DON + k_{cdom} \cdot CDOM \tag{4}$$

Terrestrial CDOM behaves nearly conservatively in the ocean. An indication is the linear salinity–CDOM relationship in the northern Baltic (Harvey et al., 2015; Neumann et al., 2020). This is due to the fact of high freshwater supply with high CDOM 65 concentrations. However, this relation does not apply to the central Baltic. In this region with longer residence time, the effects of CDOM degradation processes become more pronounced and observable (Skoog et al., 2011).

Two processes control CDOM degradation, photobleaching and biological degradation. Moran et al. (2000) study the degradation of terrestrial CDOM in the coastal ocean and find that photobleaching accounts for 80% of the degradation. Furthermore, they show that CDOM decay follows closely simple first-order kinetics. In accordance with these findings, we implement 70 CDOM as

$$\frac{\mathrm{d}CDOM}{\mathrm{d}t} = -dr \cdot CDOM \tag{5}$$

with the degradation rate

$$dr = dr_0 \cdot PAR(z). \tag{6}$$

$PAR(z)$ is the ambient PAR at depth $z$, $dr_0$ is a constant, and $PAR(z)$ can be estimated as:

$$PAR(z) = r \cdot I_0 \cdot \exp(-\int_z^0 \mathrm{d}z' K_{PAR}(z')) \tag{7}$$

$I_0$ is the solar radiation at sea surface depending on sun zenith angle, which is a function of latitude and time. Factor $r$ is the fraction of $I_0$ available as PAR (spectral range 400 to 700 nm). The integral consider the depth dependence of model concentrations of e.g. chlorophyll in Eq. 4.

### 2.1.2 Implementation of the optical model

A comprehensive overview about CDOM in the ocean is given in Nelson and Siegel (2002). CDOM concentration is usually given by a proxy, the light absorption for a specific wavelength, e.g. $a_{CDOM}(440)$ for 440 nm. The spectral distribution can be parameterized by an exponentially decline with wavelength.

$$a_{CDOM}(\lambda) = a_{CDOM}(\lambda_0) \cdot \exp(-s(\lambda - \lambda_0)) \tag{8}$$

$s$ is the exponential slope parameter and varies between $0.015 - 0.025 \, \mathrm{nm}^{-1}$. For a given slope $s$ and an absorption $a_{CDOM}(\lambda_0)$, any $a_{CDOM}(\lambda)$ can be estimated for wavelengths longer than 320 nm. We use a reference wavelength of 440 nm.

The biogeochemical framework for implementing the CDOM state variable is the model ERGOM. In this model, state variables are given as concentrations of an element, e.g. mol carbon per $\mathrm{m}^3$, because these models primarily describe cycles of elements (carbon, phosphorus, nitrogen etc.). In order to model CDOM, a relationship between CDOM absorption and the concentration is required. Following the Lambert–Beer law, a linear relation exists. Neumann et al. (2020) derived a relationship based on optical measurements and measurements with a calibrated CDOM sensor, which we used to convert CDOM absorption into concentration and vice versa. We have to note that the accuracy of the conversion does not impact the performance of the optical model because both the satellite derived CDOM in freshwater and CDOM in the optical model is given as $a_{CDOM}(440)$. CDOM loads are estimated from concentration times runoff which is available from Gustafsson et al. (2012).

Most model constants used in the model are provided by Neumann et al. (2015). We use 50% of total solar radiation for PAR (e.g., Stigebrandt and Wulff, 1987) since the invisible, long-wave part is absorbed at the water surface (factor $r$, Eq. 6). $dr_0$ (Eq. 6) has been estimated with a series of calibration simulations. Aim of the calibration was to find an optimal match of observed and simulated absorption values $a_{CDOM}(440)$. Used constants are listed in Tab. 1. Model state variables in Eq. 4 have to be converted into appropriate units before entering the optical model. We use a volume based concentration. $CDOM$ should be given as absorption because of uncertainties in the absorption–concentration relationship.

The technical implementation is done by an automatic code generation. Fundamentals are a set of text files describing the biogeochemistry independently of computer language and the host system. Code templates describe physical and numerical aspects, and are specific for a certain host e.g. a circulation model. All necessary ingredients, the code generation tool, text files, and templates for several systems, can be downloaded from www.ergom.net (last access: 22 September 2020). The same technique is used e.g. in Radtke et al. (2019).

### 2.1.3 Circulation and biogeochemical model

For model testing, we have used a similar model system as in Neumann et al. (2015). The circulation model is MOM5.1 (Griffies, 2004) adapted for the Baltic Sea. The horizontal resolution is three nautical miles. Vertically, the model is resolved

**Table 1.** Constants of the optical model

| Const. | Value | Unit |
|--------|-------|------|
| $k_w$ | 0.027 | $m^{-1}$ |
| $k_c$ | 0.029 | $m^2 \, (mg \, Chl)^{-1}$ |
| $k_{det}$ | 0.0039 | $m^2 \, (mg \, N)^{-1}$ |
| $k_{don}$ | 0.0009 | $m^2 \, (mg \, N)^{-1}$ |
| $k_{cdom}$ | 0.221 | 1 |
| $dr_0$ | 8.75e-5 | $day^{-1}$ |
| $r$ | 0.5 | 1 |

into 152 layers with a layer thickness of 0.5 m at the surface and gradually increasing with depth up to 2 m. The circulation model is coupled with a sea ice model (Winton, 2000) accounting for ice formation and drift.

Coupled with the circulation model is the biogeochemical model ERGOM. It describes a marine nitrogen and phosphorus cycle. Primary production, forced by PAR, is provided by three functional phytoplankton groups (large cells, small cells, and cyanobacteria). Chlorophyll concentration can be estimated from the phytoplankton groups which is used in the optical model. Dead particles accumulate in the detritus state variable which is another compartment in the optical model. A bulk zooplankton grazes on phytoplankton and constitutes the uppermost trophic level in the model. The metabolism of phytoplankton and
zooplankton produces DON which has only little impact on light absorption. Phytoplankton and detritus can sink down in the water column and accumulate in a sediment layer. In the water column and the sediment, detritus is mineralized into dissolved inorganic nitrogen and phosphorus. Mineralization is controlled by temperature and oxygen. Oxygen is produced by primary production and consumed due to all other processes e.g. metabolism and mineralization. Coupled to the nitrogen and phosphorus cycle is a carbon cycle as described in Kuznetsov and Neumann (2013). A schematic of the model structure is
provided in Appendix B. The estimated short wave absorption (Sect. 2.1.2) feeds back into the physical part of the model and hence impacts the temperature distribution.

    The new CDOM variable in the current model development state is not involved in the biogeochemical processes. This is justified by the fact that CDOM is relatively refractory and has a long residence time, and autochthonous CDOM produced by e.g. phytoplankton, is a small fraction. In later developments, it will be included in the carbon cycle. For this purpose, it
is essential to realize CDOM as a carbon based concentration. If this will be not the case, the CDOM state variable could be implemented as an absorption and thus conversions between absorption and concentration could be prevented.

    The model has been forced by meteorological data from the coastDat-2 data set (Geyer and Rockel, 2013). We run the model from 1948—2019. A first run was used to spin up the new CDOM tracer. In a second run, CDOM was initialized with data from the first run. In addition to the 3 nautical miles resolution, we use a 1 nautical mile resolution for the period 2017–2019.
The model has been successfully used in several applications (e.g., Neumann, 2010; Neumann et al., 2015).

## 2.2 CDOM boundary data from Earth Observation products

Aim of the development is to improve the simulated light climate by a more realistic representation of CDOM concentration compared to available statistical models (Sect. 1). However, *in situ* data of CDOM loads are not available in sufficient spatial and temporal resolution, i.e. in the Baltic Sea, CDOM is not a parameter of the HELCOM (www.helcom.fi, last access: 22 September 2020) monitoring program. New instruments and technologies in Earth Observation (EO), now in operation, are ideal tools to overcome these limitations.

### 2.2.1 Characteristics of satellite data

In order to estimate the load of CDOM coming from a river, it is necessary to derive it from observations within the river or as close to the discharge point as possible. The rivers in the Baltic Sea are usually small and thus high resolution (HR) instruments such as Sentinel-2 Multi Spectral Instrument (S2-MSI) are required. The MSI has a spatial resolution of 10-60 m depending on the central wavelength of the band. Water quality products are usually generated in 60 m resolution in order to reduce noise. This is sufficient for estimating the CDOM absorption of most rivers.

Two Sentinel-2 satellites are currently in orbit: S2A was launched on 23 June 2015 and S2B on 7 March 2017. Together they provide a global revisit time of 5 days next to equator. Due to the high latitude of the Baltic Sea, the revisit time amounts to 2–3 days in this area. Despite the frequent cloud cover, sufficient observations can be gathered to monitor river CDOM throughout the open water season (typically from March-April to October in the northern Baltic Sea).

### 2.2.2 Earth Observation processor for CDOM absorption estimation

Earth Observation (EO) processors are a set of algorithms designed to convert the radiance signal acquired by the satellites into values of geophysical parameters. The estimation is based on the scattering and absorption features of the material suspended or dissolved in water. In addition to CDOM, these materials include phytoplankton cells, represented by Chlorophyll a (Chl-a), and suspended particulate matter.

One commonly used water quality processor is Case 2 Regional Coast Color (C2RCC) (Brockmann et al., 2016). It utilizes one artificial neural network (ANN) first to remove the effects of the atmosphere from the signal (atmospheric correction) and then another to estimate inherent optical properties (IOPs) of water from the marine reflectance.

For estimation of CDOM absorption ($a_{CDOM}$), we utilized the C2RCC (version 1) output called $a_{dg}$ (combined absorption by detritus and yellow substances) which was calibrated to $a_{CDOM}(440)$ values (absorption coefficient by CDOM at 440 nm) using this equation:

$$a_{CDOM}(440) = 0.654 \cdot a_{dg}^{1.45} + 0.2 \tag{9}$$

The equation is based on in situ sampling made with a flow-through device (ac-9) (Lindfors et al., 2005; Koponen et al., 2007) during two coastal estuary measurement campaigns. These data are not yet published but a similar local calibration method has provided good results with other water quality parameters in earlier studies such as Attila et al. (2013).

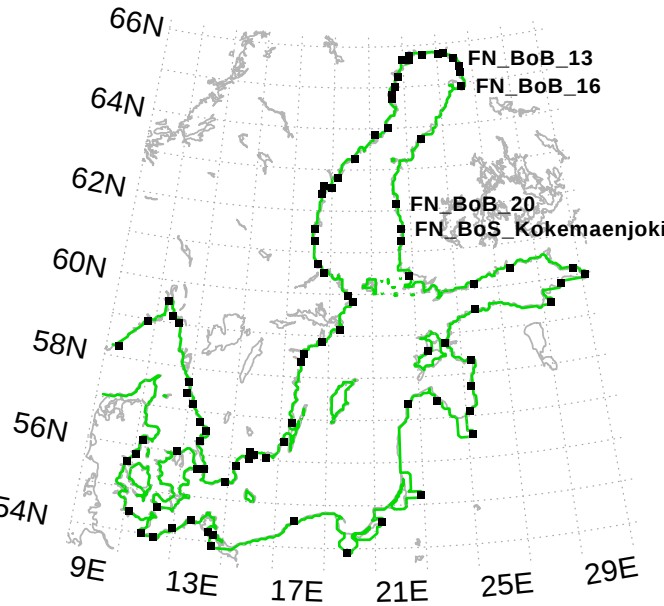

**Figure 1.** Location of model rivers (black squares). The green line is the coastline of the 3 nm model. We refer to the labeled river later in the text. The map was created using the software package GrADS 2.1.1.b0 (http://cola.gmu.edu/grads/), using published bathymetry data (Seifert et al., 2008)

The data processing and extraction are done in a Calvalus massive parallel processing system (http://www.brockmann-consult.de/calvalus, last access: 22 September 2020). Data extraction areas are manually defined in the vicinity of the mouths of 69 rivers that represent ERGOM input locations (Fig. 1). The areas are designed so that islands, mixed pixels and shallow areas are excluded. All valid pixels (not masked as land or cloud by the pixel classification processor Idepix) within each area and image are collected and analyzed, and the 75th percentile value is chosen to represent the river $a_{CDOM}$. The cases in which the number of valid pixels is less than 50% of all available pixels from an area are removed from the analysis. Assumedly, these represent cases with partial cloud cover and they are discarded to keep only estimates with highest quality and low uncertainty. The arithmetic means of the 75th percentile pixel values of all valid days within each calendar month during years 2017–2019 are then computed for each extraction area.

We are aiming at providing the ecosystem model ERGOM with an annual cycle of CDOM loads based on monthly data. Since optical EO methods cannot provide $a_{CDOM}$ estimates in darkness and throughout times with ice coverage, the values for the winter months have been interpolated. As a result, the dataset contains $a_{CDOM}$ value for each month for each of the areas under investigation (69 extraction areas in total). Figure 2 shows four examples of the annual CDOM absorption cycle. The behavior of the data follows well the annual cycle: spring values are high due to the terrestrial matter brought into the coastal water by melting snow, summer values are low due to lower influx and the fall values are higher due to increasing rainfall.

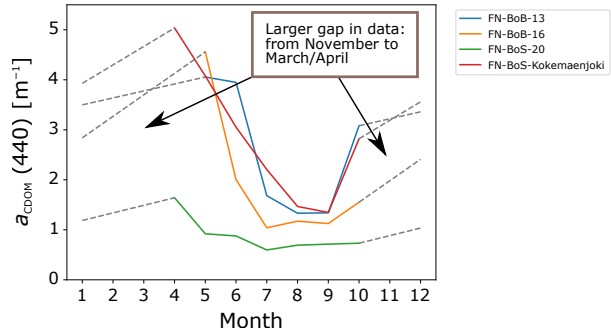

**Figure 2.** The aggregated monthly values of $a_{CDOM}$ based on EO data (S2 & C2RCC V1) for four ERGOM input locations in the western coast of Finland. Location of the rivers is shown in Fig. 1.

## 3 Results and discussion

In this section, we show the improved model CDOM representation and its impact on the simulation results. Owing to the changed shortwave distribution in the water column, an effect especially on the biogeochemistry is expected. All CDOM data presented are converted into absorption at $440\,\mathrm{nm}$ ($a_{CDOM}(440)$). Especially for observations at different wavelengths, we use Eq. 8 with a slope $s$ of $0.018\,\mathrm{nm}^{-1}$ (Kratzer and Moore, 2018). For comparison, we show $a_{CDOM}(440)$ values estimated from the CDOM state variable and from model salinity. The models differ only in the estimation of PAR which becomes evident when we show the impact on biogeochemistry.

### 3.1 CDOM absorption

In Fig. 3a, we show the simulated CDOM absorption at the sea surface. The snapshot clearly illustrates the spatial patterns. Strong absorption is visible in the northern Baltic and the river mouths. The difference to the salinity based estimate (Eq. 2) is depicted in the right panel. Strongest differences appear in the Gulf of Bothnia and the Gulf of Finland while in the central Baltic differences are small. Strong differences are also pronounced in river estuaries. Owing to the low salinity, the salt-CDOM relationship overestimates CDOM content in these areas. Furthermore, the new, EO based method considers individual CDOM concentrations of different rivers. Rivers of the northern catchment area carry higher CDOM loads compared to rivers of the south-eastern catchment area due to a high fraction of peat land. The range of $a_{CDOM}$ values is demonstrated in Fig. 4. Both datasets were compared against *in situ* data collected from monitoring stations in coastal waters of Finland and from the Northern coast of Sweden. As shown in Fig. 5, the improvement becomes obvious. With the salinity method, the correlation is low ($\mathrm{R}^2 = 0.15$) and there are some clear overestimates (difference between $a_{CDOM}(440)$ from salinity and the one-to-one line more than 2 $\mathrm{m}^{-1}$) while most data points are underestimated. With the EO CDOM method, the correlation improves significantly ($\mathrm{R}^2 = 0.61$). There are no large overestimates and the data points move closer to the one-to-one line. Large *in situ*

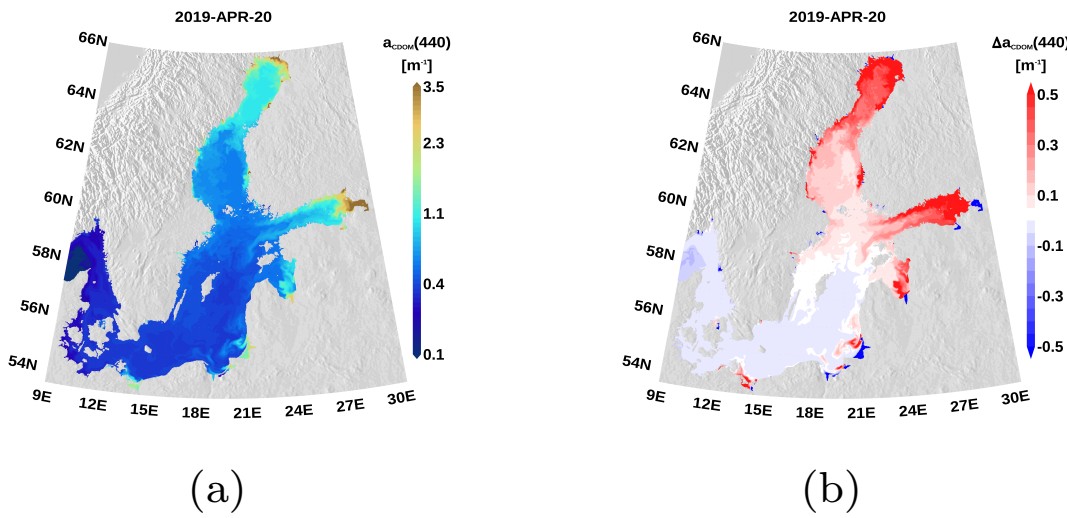

**Figure 3.** Snapshot of simulated surface $a_{CDOM}(440)$ at April 20th 2019 (a) and the difference to the salinity based absorption estimate (b as seen in the 1nm resolution model.

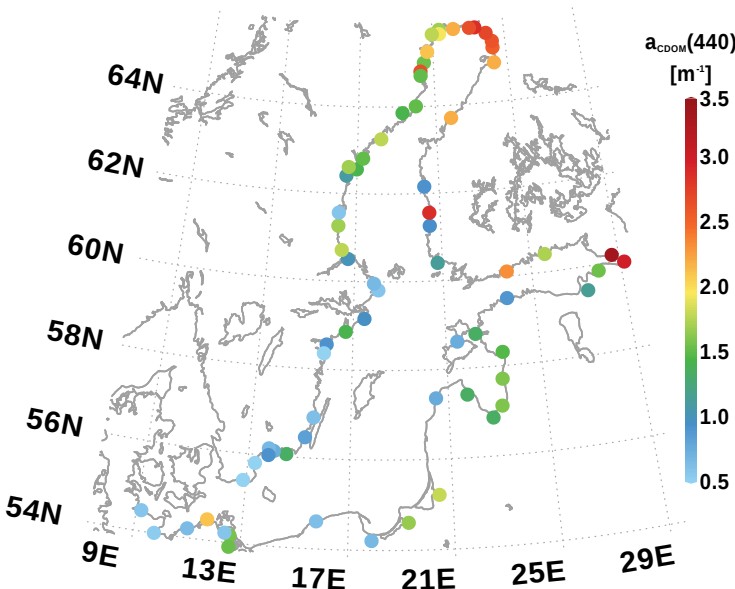

**Figure 4.** Mean $a_{CDOM}$ of the individual rivers used in the model. The map was created using the software package GrADS 2.1.1.b0 (http://cola.gmu.edu/grads/), using published bathymetry data (Seifert et al., 2008).

values ($a_{CDOM}(440) > 3 \text{ m}^{-1}$) are still underestimated with the new EO method. This underestimation is most likely caused by the following two major inaccuracies in the present version of the model input:

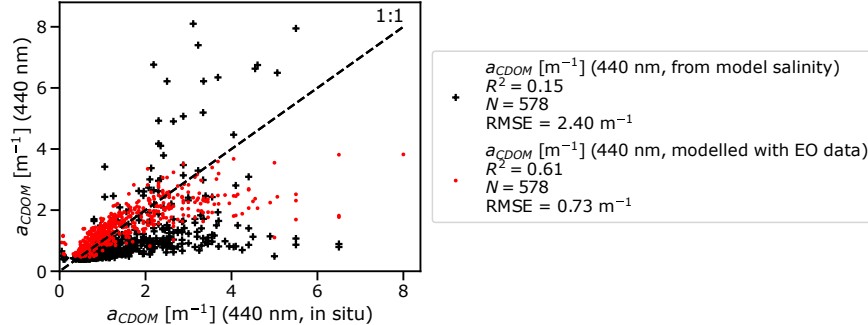

**Figure 5.** *In situ* $a_{CDOM}$ (x-axis) vs. ERGOM $a_{CDOM}$ (y-axis) estimated from salinity (in black) and ERGOM $a_{CDOM}$ with the EO method (in red).

– The coast has many small rivers. Not all of them are yet included in this version.

– In river estuaries with low bottom depth or complex morphological structure, the shapes and formulations of the extraction areas do not sufficiently capture the incoming CDOM loading from the river. In order to avoid EO observations contaminated by bottom reflectance it was necessary to use pixels that are sufficiently far from the shoreline. Therefore, pixels of the extraction area may not represent river water as it has been already mixed with sea water. In some cases, this leads to lower concentrations especially during the low runoff season.

Figure 6 demonstrates the different CDOM absorption estimates. Shown are time series of surface CDOM absorption at 6 stations (Fig. 6d). The green curve is the salinity based estimate and the black curve the estimate from model CDOM concentration. Red diamonds are *in situ* observations.

At stations 1–4, absorption values from simulated CDOM are much closer to observations compared with salt based estimates. The absorption difference at stations 5 and 6 is less pronounced which is also evident from Fig. 3. The seasonal 210 variability is stronger for absorption derived from the model CDOM variable (compared to salt based absorption) and reflects the observed variability. The reason is the annual cycle of riverine CDOM concentration in addition to the runoff cycle. In the central Baltic Sea, both methods overestimate CDOM absorption.

### 3.2 Impact on biogeochemistry

As an example of the changed light absorption impact on the biogeochemistry, we show annual mean profiles of selected 215 variables in Fig. 7. We have chosen station 4 from Fig. 3 since at this station the CDOM absorption is considerably increased due to the new, EO based, optical model and it is located in the center of the Bothnian Bay. Owing to the increased CDOM absorption, PAR is reduced in the EO model approach (Fig. 7a) as expected. Consequently, primary production (PP) is reduced (Fig. 7b). However, in the uppermost layer, PP is increased. As a result of reduced PP, phytoplankton concentration shows lower values (Fig. 7c). An integrated response is the increased bottom oxygen concentration (Fig. 7d). Less net PP results in

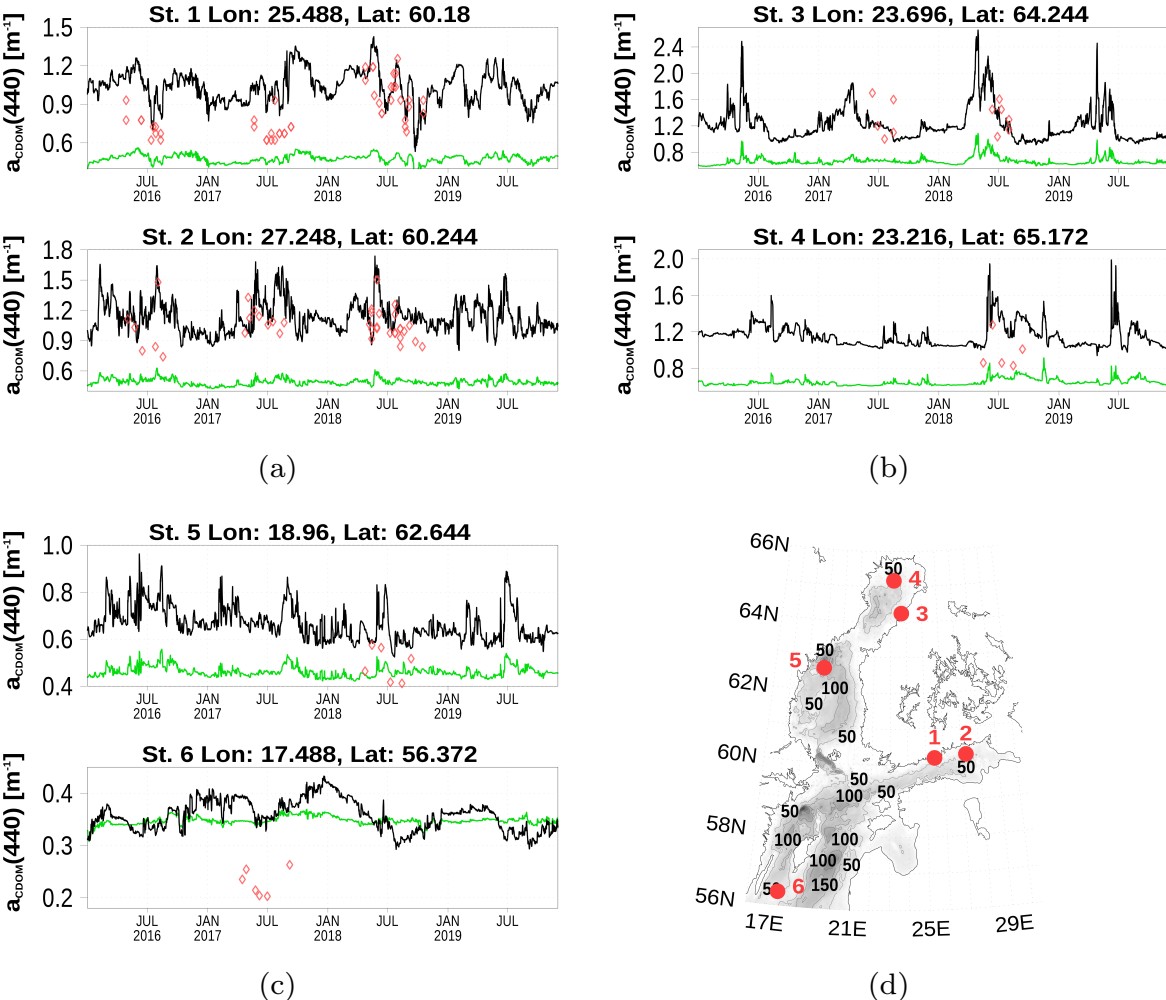

**Figure 6.** Surface $a_{CDOM}(440)$ time series at 6 stations. Location of the stations is shown in (d). Absorption estimates based on simulated CDOM are shown in black, based on a conversion from simulated salinity in green, and red diamonds are observations. The map was created using the software package GrADS 2.1.1.b0 (http://cola.gmu.edu/grads/), using published bathymetry data (Seifert et al., 2008).

less accumulation of organic matter in the deep water of the basin and subsequently reduced oxygen consumption. The impact on water temperature is small (order of 0.01 K, not shown). The effect is a temperature increase in the surface layer and a lower temperature below.

We demonstrate changes in the biogeochemistry with a climatology of surface nutrient concentrations at three stations in Fig. 8. For this analysis, we use data from the 3 nautical miles model version because of the longer simulation period. Shown are data from the EO model (black) and the previous (salinity based) model (green) together with observations (red). In the Gulf of Finland at station KAS-11, the spring bloom related nutrient depletion is delayed by 2 weeks (Fig. 8a and b). Sufficient

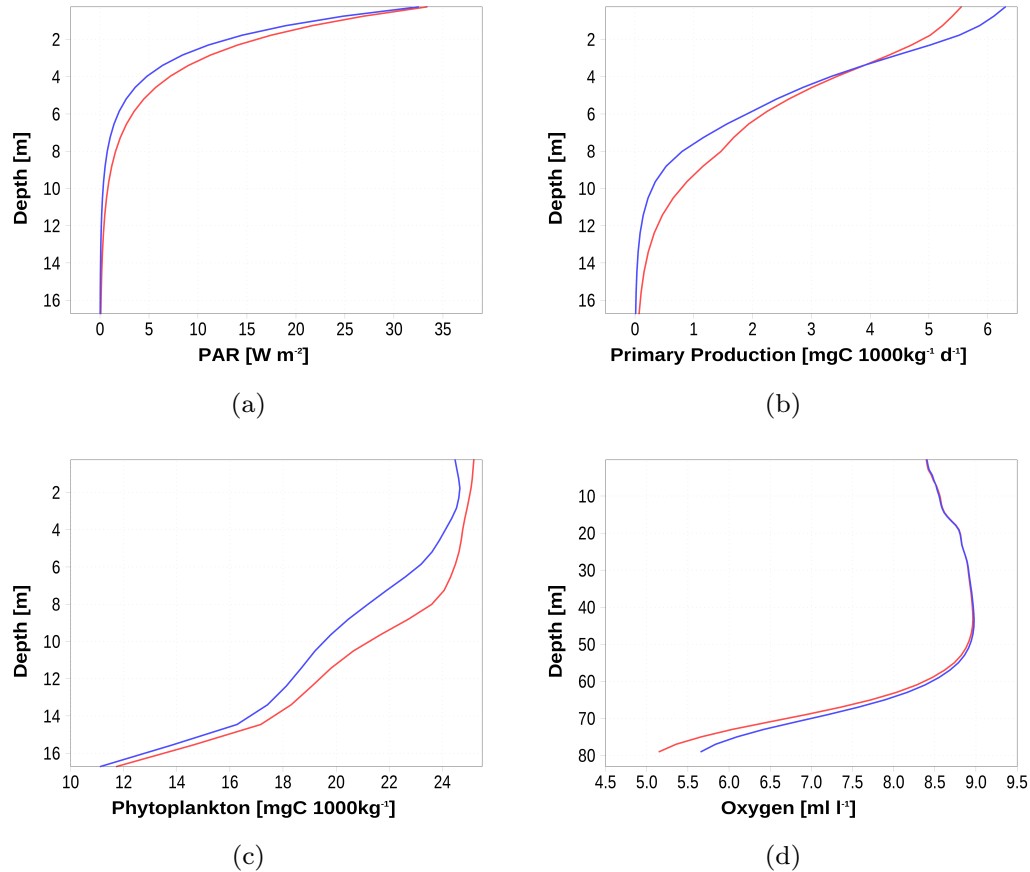

**Figure 7.** 2018 annual mean profiles at station 4 (Fig. 6d). Blue curve shows data from the model with an explicit CDOM state variable and the red curve is from the model with CDOM absorption estimates based on its relation to salinity. For oxygen (d), we show the whole water column.

PAR intensity, initiating a bloom, is available later in the season. The winter nutrient concentrations are elevated compared to the salinity model version. At the Bothnian Bay station (Fig. 8c and d), the spring bloom delay is less pronounced. In this area, the longer sea ice coverage dominates the PAR in spring. At station BY15 in the Baltic Proper (Fig. 8e and f), the difference

between both model versions is small due to similar CDOM absorption (Fig. 6).

## 4   Summary and conclusions

In this study, we propose an approach for considering light absorption due to terrestrial CDOM in a marine ecosystem model for the Baltic Sea. An explicit consideration is necessary if large amounts of terrestrial CDOM enter the marine system and strong coastal-sea gradients develop. In such cases, a uniform light absorption due to CDOM cannot account for the *in situ*

light climate in a sufficient way. An often applied approach uses CDOM-salinity relationships for CDOM absorption estimates (Kowalczuk et al., 2006, 2010; Neumann et al., 2015) but with distinct disadvantages (see Sect. 1).

Our approach uses an explicit CDOM state variable as part of the biogeochemical model. In order to improve the simulated absorption compared to the salt approximation, a high quality data set of riverine CDOM loads is necessary. This has been accomplished by using earth observation data from Sentinel-2 MSI. The high spatial resolution (10 m–60 m) allows to observe

the river mouths directly. A difficulty in the regions of higher latitude like the Baltic Sea area is the insolation, the occurrence of sea ice, and the frequent cloud cover in winter. Continuous observations are not possible during this time. We have used a linear interpolation to bridge the winter data gap. This could be validated by ground truth measurements in winter possibly guiding for another than linear interpolation.

The results (Sect. 3) show that the proposed approach clearly improves the ability of the model to estimate CDOM and

245 thus light absorption especially in the northern parts of the Baltic Sea where the impacts of terrestrial CDOM are large. This underlines the performance of the combined approach to increase the predictive capability of ecosystem models. The method can be further improved by adding more rivers to the model and improving the quality of CDOM data from Sentinel 2 MSI.

For the model CDOM, we have applied a light sensitive degradation. Although this is the dominating degradation process for terrestrial CDOM (Moran et al., 2000), bacterial breakdown contributes to the degradation as well. Technically, such a

250 process can easily be implemented. However, to our knowledge comprehensive process studies in the Baltic Sea are not done yet. Therefore, we have decided that bacterial breakdown is subject to later developments.

We consider only terrestrial CDOM in our model. In regions with high runoff, like the Baltic Sea, terrestrial CDOM is the dominating fraction (Harvey et al., 2015; Stedmon and Markager, 2003). However, a further step toward a more sophisticated model could be the inclusion of autochthonous CDOM as e.g. in Dutkiewicz et al. (2015).

*Code and data availability. In situ* absorption observations are available from http://eo.ymparisto.fi/data/water/Baltic_SeaLaBio/. Monthly CDOM absortion data are available from http://eo.ymparisto.fi/data/water/Baltic_SeaLaBio/CDOM_input_to_ERGOM/. Model data can be accessed via https://thredds-iow.io-warnemuende.de/thredds/catalogs/projects/SeaLaBio/catalog_sealabio.html.

The code of the biogeochemical model is available at www.ergom.net (last access: 22 September 2020). The ocean model "Modular Ocean Model MOM 5-1", used in this study, is available from the developers respository https://github.com/mom-ocean/MOM5 (last ac-

260 cess: 1 December 2020). The meteorological forcing is archived at https://cera-www.dkrz.de/WDCC/ui/cerasearch/entry?acronym=coastDat-2_COSMO-CLM (last access: 1 Decenber 2020). The version of the model code used to produce the results in this study is archived on Zenodo at https://doi.org/10.5281/zenodo.4299873 (last access: 1 December 2020). In addition to the source code, the archive includes initial fields and boundary conditions exept the meteorological forcing.

Nitrate and phosphate date used for model comparision are available from the ICES database https://ocean.ices.dk/Helcom/Helcom.aspx?Mode=1

(last access: 30 Aril 2021).

*Sample availability.* Simulated CDOM data: https://wwwi4.ymparisto.fi/i4/eng/tarkka_beta/index.html?type=ERGOM_CDOM&date=2019-12-01&lang=en&zoom=5&lat=61.46508&lon=32.98851

## Appendix A: Impact of photobleaching

Photobleaching accounts for slow decomposition of CDOM. Although, the CDOM decomposition is slow compared to de-
composition of e.g. *in situ* detritus, in water bodies with longer residence time it becomes important. For example, in the salt –
CDOM absorption relationship, decomposition is considered by a hyperbolic function (Neumann et al., 2015, eq. 10):

$$a_{CDOM} = 1.26\, S^{-0.627}\, [m^{-1}] \tag{A1}$$

We demonstrate the effect of the implemented photobleaching on the model CDOM concentration by comparing a simulation
without photobleaching with the control run. The 3 nautical miles setup was used for this study. The photobleaching was
switched of in 1980 and then the simulation was continued until 2019. Figure A1 shows the differences developing due to
lacking a degradation process. Absorption values are far away from observations and even after 40 simulation years, a new
steady state is not achieved (Fig. A1a). Spatial patterns after 39 simulation years show that largest differences occur in the
central basins. Differences are smaller in freshwater dominated regions (Fig. A1b) like river estuaries. Small differences are
also in the vicinity of the open boundary toward the North Sea.

## Appendix B: Simplified schematic of the biogeochemical model ERGOM

In Fig. B1, we show the structure of the biogeochemical model ERGOM. Ellipses are state variables and rectangles are pro-
cesses describing the transfer from one to another state variable. The meaning of the state variable symbols are given in
Table B1.

*Author contributions.* TN developed the ecosystem model components. SK led the project. SV performed the processing of EO data. PY
performed CDOM absorption measurements in the central Baltic Sea. All authors designed the project, and contributed to data analysis and
writing the manuscript.

*Competing interests.* The authors declare that they have no conflict of interest.

*Acknowledgements.* This work was supported by ESA Contract No. 40000126233/18/I-BG (BALTIC+ SeaLaBio). Computational power
was provided by the North-German Supercomputing Alliance (HLRN).

**Table B1.** State variables of the biogeochemical model ERGOM

| Symbol | State Variable |
|---|---|
| $O_2$ | dissolved oxygen |
| $N_2$ | dissolved nitrogen |
| CDOM | colored dissolved organic matter |
| DIC | dissolved inorganic carbon |
| TA | total alkalinity |
| $NH_4$ | ammonium |
| $NO_3$ | nitrate |
| $PO_4$ | phosphate |
| $SO_4$ | sulfate |
| S | sulfur |
| $H_2S$ | hydrogen sulfide |
| large cells | large cell phytoplankton |
| small cells | small cell phytoplankton |
| cyanobacteria | cyanobacteria |
| zooplankton | bulk zooplankton |
| detritus | detritus |
| DOC | dissolved organic carbon |
| $DOC - N$ | DOC with additional nitrogen |
| $DOC - P$ | DOC with additional phosphorus |
| POC | particulate organic carbon |
| $POC - N$ | POC with additional nitrogen |
| $POC - P$ | PC with additional phosphorus |
| sediment detritus | detritus accumulated in the sediment layer |
| $Fe(III) - PO_4$ | iron-3-phosphate |

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

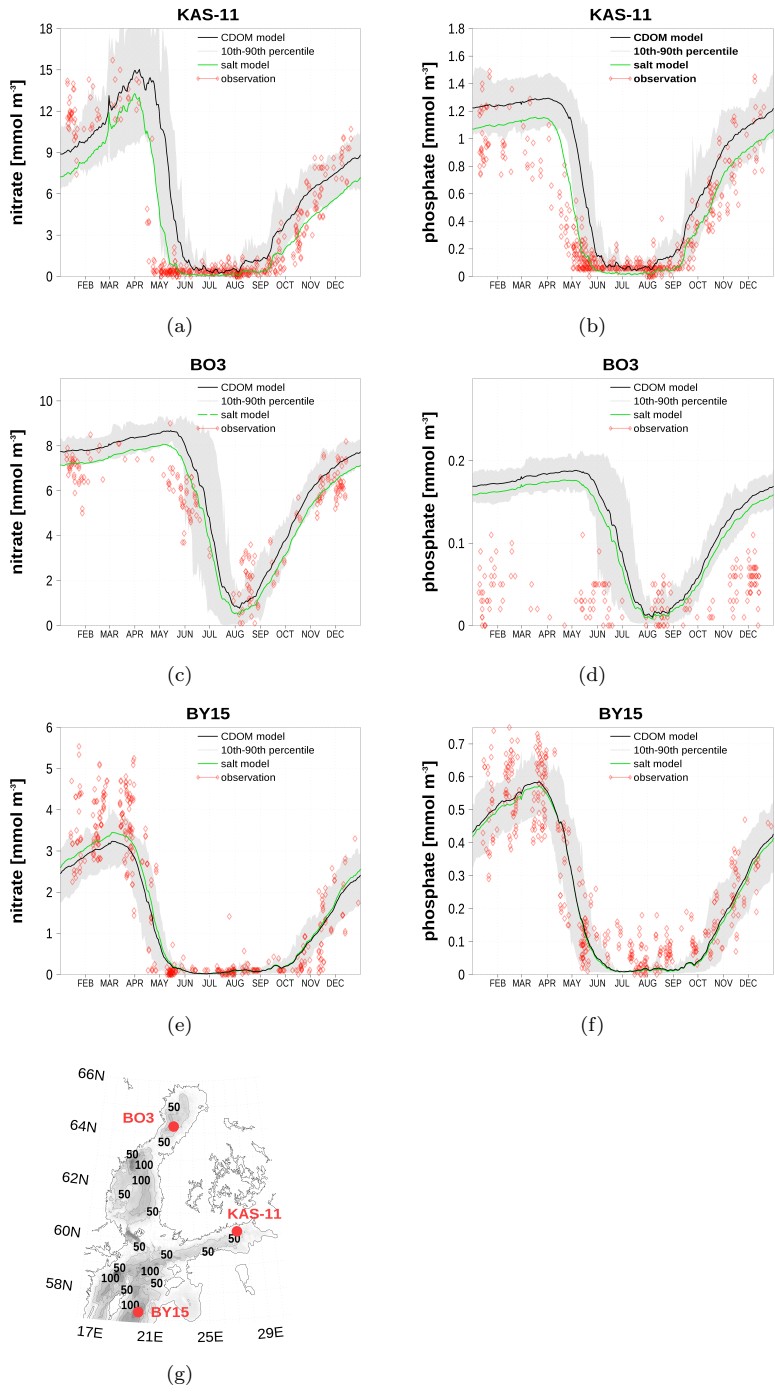

**Figure 8.** Climatology (1990–2019) of surface nitrate and surface phosphate at 3 stations. Black lines show the new CDOM based model, green line the salt based model, and red diamonds are observation (http://www.ices.dk, last access: 19 June 2020). The shaded area is the range between 10th and 90th percentile of the black line. Simulated data are from the 3 n.m. model version. The map was created using the software package GrADS 2.1.1.b0 (http://cola.gmu.edu/grads/), using published bathymetry data (Seifert et al., 2008).

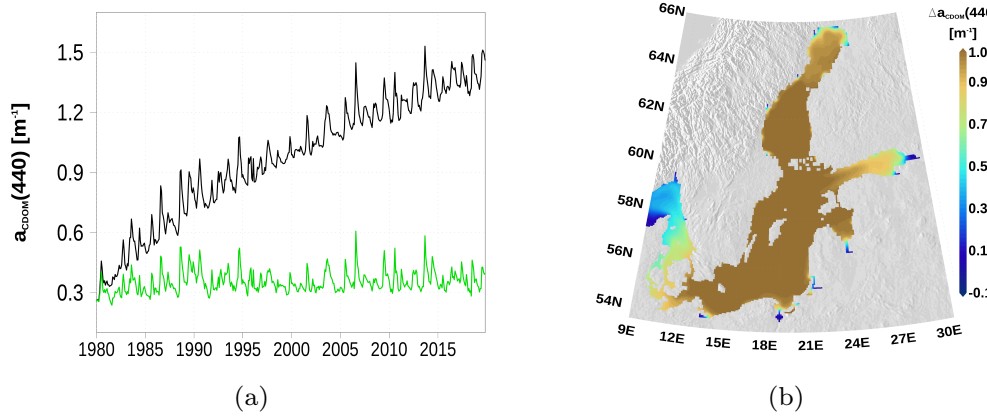

(a)              (b)

**Figure A1.** The effect of photobleaching on CDOM absorption. The black line in (a) is from the model without photobleaching at station BY15 (Fig. 8). In (b) the difference in 2018 of the surface CDOM absorption is shown (without photobleaching minus control run). Simulated CDOM concentration have been converted into absorption. The map was created using the software package GrADS 2.1.1.b0 (http://cola.gmu.edu/grads/), using published bathymetry data (Seifert et al., 2008).

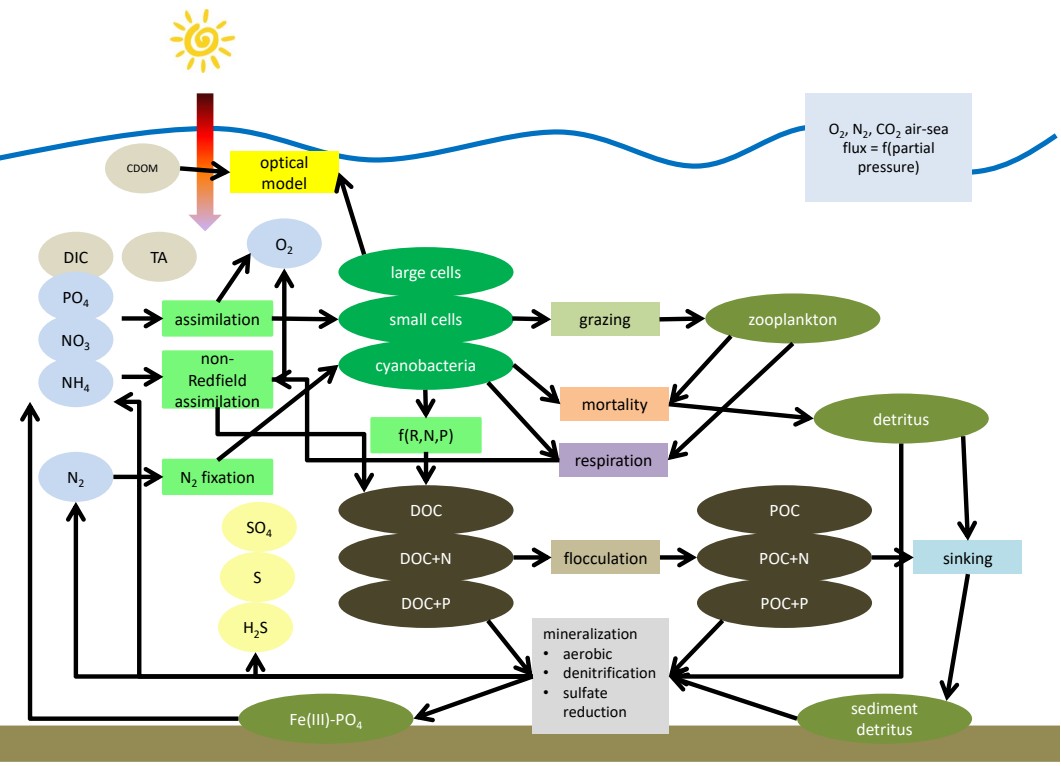

**Figure B1.** Simplified schematic of the biogeochemical model ERGOM. State variables are shown as ellipses and processes as rectangles.