# Peer review of "Optical model for the Baltic Sea with an explicit CDOM state variable: a case study with Model ERGOM (version 1.2)"

_Geoscientific Model Development, 2020_

## Referee Comment (RC1) · M. Baird (Referee) · 12 Mar 2021

This paper uses satellite determinations of PAR-weighted CDOM absorption in the mouths of 69 estuaries / rivers as boundary conditions for a model tracer that represents the absorption of CDOM in a Baltic Sea configuration of the Modular Ocean Model (MOM). The authors find the absorption calculated with the new formulation improves on a simple CDOM-salinity relationship when compared to in situ absorption measurements, and it is shown that this change affects a biogeochemical model.

I found this an excellent and thought-provoking approach to take which will be of interest to coastal modellers. The manuscript itself has a significant number of problems

which I discuss below, and I have a few insights that might simplify / improve the approach.

Major comments.

1. The authors are occasionally loose with the use of CDOM vs. CDOM absorption and this becomes confusing. For example, p2 L17 is CDOM absorption; L26 'amount of CDOM'. Is this a load, concentration, rate of absorption?

2. While it is not entirely clear, the authors appear to attribute the improvement of their CDOM equation compared to the salinity relationship to the inclusion of non-conservative behaviour CDOM. Instead, I suspect the majority of the improvement is due to the use of the 65 stations to better set the inflow concentrations of CDOM. If they run a simulation with the non-conservative terms set to zero they would be able to quantitatively compare the importance of one over the other. The comparison should also include a complete description of the salinity-CDOM absorption parameterisation so that we understand the comparison. For researchers such as myself considering both options in a coastal model, this would double the value of the paper.

3. The authors use a neural network to determine the CDOM component of absorption at 440 nm at 65 sites. This is a key innovation. They then undertake a convoluted set of calculations, including choosing an arbitrary 75th percentile value, in order to turn the satellite-determined absorption into a CDOM concentration which is then multiplied by kcdom in Eq. 5 to obtain the component of vertical attenuation due to absorption CDOM. Is this complicated pathway even necessary? Absorption is inherently additive. Furthermore, the degradation rate is proportional to concentration, which is itself proportional to absorption. So, could you not simply have a model tracer CDOM absorption at 440, and applying the mixing and non-conservative terms to this tracer?

4. I presume the use of the 75th percentile is about trying to determine the CDOM absorption in the freshwater end member. Given that there is a hydrodynamic model, the authors could use salinity in the hydrodynamic model to determine the unique freshwater endmember for each of the 65 sites? I know you started with the feeling that salinity vs CDOM doesn't work, but I think this is because the Baltic has 65 different freshwater end members.

5. One of the key findings is the variability in CDOM absorption at the 65 stations. Fig. 4 illustrates this, but much more could be shown. I suggest splitting the two panels into 2 figures, and showing the map as large as Fig. 1, but with the 65 sites with symbols collected by mean absorption at 440. For some researchers this alone would be an important result.

Minor comments.

Title: "Radiation model" might imply a more sophisticated, directional model of light. Perhaps "Optical model" is less specific?

P1 L17 what does divergence mean in this context?

P2 L27 always have a space between a quantity and its units.

P2 2nd para. Paragraph goes from discussing non-conservative behaviour (2nd sentence), conservative behaviour (4th) to non-conservative again (5th). I understand what you are trying to say, and of course the point of the paper is in part the non-conservative behaviour. Paragraph just needs a more logical flow.

P3 EO processors – does this mean software, theory?

P3 L9 water leaving reflectance is a tautology.

P3 L12 coastal waters of Finland

P5 L15 'behaves conservatively',

P5 Eq. 8. Replace '2' in the equation with a parameter, the fraction of SWR in total solar radiation. What is the difference between PAR(z) and I(z)? Are you sure about exponential term in Eq 8. The K_PAR in front of the integral doesn't seem write. For

equations, paper "Edwards, A.M. and Auger‐Méthé, M., 2019. Some guidance on using mathematical notation in ecology. Methods in Ecology and Evolution, 10(1), pp.92-99." Is helpful.

P5 L10 K_CDOM is a parameter, not a statistical relationship.

P5 L10 CDOM absorption?

P5 L14. Isn't DON part of the DOM? In which case are the last two terms in Eq. 5 double counting?

P5, 2nd last line. 'depending on sun zenith angle, which is a function of latitude and time of day"

P6 L8 per m3

P6 L22 Whet is "Basis"?

P7 Title 3.3 Model configuration?

P7 A schematic of the biogeochemical model would help here.

Fig. 3b colorbar caption should be delta a(440).

P9 First paragraph – this discussion needs to be more quantitative.

---

## Referee Comment (RC2) · Svetlana Losa (Referee) · 6 Apr 2021

Review on „Radiation model for the Baltic Sea with an explicit CDOM state variable: a case study with Model ERGOM (version 1.2)" by T. Neumann et al.

The study is dedicated to augmenting the MOM-ERGOM-based model system of the Baltic Sea by accounting of the effect of the light absorption by coloured dissolved organic matter (CDOM) when modelling the light path with implication on the biogeochemistry of the basin. Moreover, to simulate more precisely the light attenuation due to CDOM, the authors consider this optical constituent as an additional model variable, that has not yet been incorporated in the biogeochemical cycling. Neither the authors consider a part of CDOM produced as a result of phytoplankton functioning, but rather first as a tracer of a terrestrial origin, which is, nevertheless, crucial - and makes such an implementation valuable, - for the investigated basin. In this respect the study suits the frame of the journal. The strongest part of the presented study is the exploiting a satellite CDOM data product for representation of the CDOM loading by rivers (for specification of model CDOM boundary conditions). My comments mostly concern the manuscript's structure (clarity) and quality of the figures, which needs to be improved. Below I listed a number of comments/suggestions the authors might want to consider and address in a revised version of the manuscript. After such a revision I would recommend the paper for publishing.

**General comments:**
1) An edit is required for the title (please follow recommendation of reviewer 1)
2) In the abstract, please present more precisely the evaluation results (including comparison of "the traditional" approach). How exactly did the model performance with the new light attenuation parameterisation improve, given which particular evaluation criteria?
3) Introduction should be extended more intensively by references to the state-of-the-art of the investigated problem and related studies (see my specific comments), which would show the present study in line with already existing research and would further emphasise the added value.
4) The manuscript could benefit from a restructuring. In particular,
Part 2**:** I would suggest to introduce/organize a separate section: **2 Methods and data** and started first (Section 2.1) with model description
   - general (MOM-ERGOM) model description
   - Radiation (optical) model development
      o   Implementation in ERGOM
followed by
   - data description (Section 2.2) including data processor *etc.* to prescribe required boundary conditions;
   -   and further details on the experiment set up including forcing and initial conditions
and further followed by validation/evaluation metrics (Section 2.3)

5) Generally, I would also recommend elaborate a bit more on the results (however I do not list specific comments with respect, except for a request on quantitative estimates of the discussed correlations).

**Specific comments:**

P1. L13-16: It would be nice to support your statements by related references (sentence-wise).

P1. L17: Please add related references in support to the statement (*"Water temperature is affected by CDOM absorption as well"*). For instance:
Hill, 2008; Kim et al, 2015; Kim et al., 2018; Gnanadesikan et al., 2019, Soppa et al., 2019, Pefanis et al. 2020.

L19: Provide related references

P2. L2: Even for open ocean several studies showed a better representation of the light path when explicitly accounting for light absorption by chlorophyll and CDOM (Kim et al. 2015, Kim et. 2016, Groeskamp&Iudicone, 2018, Pefanis et al., 2020). Nevertheless, I agree that for coastal ecosystem it is extremely crucial (Cahill et al. 2008; Jolliff&Smith, 2014; Juhls et al. 2020).

P2. L3: I would suggest "parameterisation of light (penetration)" instead of "parametrization of model"

P2. L4: "autochthonously" instead of "autochthonous"

P2. L14-15: the authors might want to add the following references:
 Dutrkiewicz et al. 2015, Pefanis et al. 2020

P2. L15: "**In relation to the Baltic Sea,** a necessary prerequisite …."

P2. L19: A rephase is required:  "... we discuss the effect of the new development (proposed model extension?) on the ... Baltic Sea ... "

P2. L21: please consider editing of this sentence.

P3. L7-9: An edit is required for this sentence. As an example:
"It utilizes an artificial neural network (ANN) first to remove ... and then to estimate ... "

P3. L11: "gelbstoft" - please use English term :-)

P3. L11: "440 nm" (space in between)

P3. L12: "measurements from Finland" - please provide a related reference and/or link

P3. L15: "… **by** Koponen et al. (2007) **and** Attila et al. (2013)."

P3. L20: "The cases…" instead of "Cases"

P5. L2: I would suggest: "as in the study by Neumann et al. "
instead of "proposed by"

P5. L17-18: please provide a supporting reference to the statement.

P5. Equation 8: please edit the integral part of the equation.

P6. L16-21: consider combining the corresponding text in one paragraph.

P7. L7: "0.5 m" and "2 m" instead of "0.5m" and "2m"

P7. L19: CDOM as a product of phytoplankton is neither considered. Right?

**Part 4**: since there is no a discussion part, the best title would be "Results and discussion" (not just "Results")

P7. L27-28 (second sentence of Part 4): strictly speaking it might also impact the physics, but probably not in the current set up… Somehow, it was not clear enough from Parts 2/3 if the authors consider CDOM effect on the shortwave radiation penetration (and related physical processes) in general or only as a role of CDOM absorption in attenuation of the light available for phytoplankton production/growth. Please provide required emphasises.

P7. L28-32: these sentences should belong to the "Method" part.

P9. L4-3: "the correlation is low" – please provide quantitative estimates if possible.

P9. L4-5: "correlation improves …" – provide the quantitative estimates (r = …)

P9. L13-20: The text belongs to one (joint) paragraph.

P9. L31-32: might belong to the "Method"

**Part 5 Conclusions:** reads rather as "Summary and conclusions"

P10. L7: editing is required for "an approach for light absorption"
As a suggestion: "...an approach for accounting for the light absorption due to ..."
Or "...an approach for approximating/considering the light absorption due to ..."

P10. L10: "A common approach uses CDOM-salinity relationship for …"
Readds too general, please rephrase, since not all studies in existence use CDOM-salinity relationships to represent CDOM in models.

P12, L5: the authors might want to refer to the study by Dutkiewicz et al. (2015).

**Figures**

Figures 2, 4: to improve the quality of the figure please increase the size of the font used.

Figure 3: increase the size of the figure panels.

Figure 6 caption: "… based on its relation to salinity" instead of "based on salt."

**References:**

Cahill, S. O., Chant, R., Wilkin, J., Hunter, E., Glenn, S., & Bissett, P. (2008). Dynamics of turbid buoyant plumes and the feedbacks on near-shore biogeochemistry and physics. *Geophysical Research Letters*, 35, L10605. https://doi.org/10.1029/2008GL033595

Dutkiewicz, S., Hickman, A. E., Jahn, O., Gregg, W. W., Mouw, C. B., & Follows, M. J. (2015). Capturing optically important constituents and properties in a marine biogeochemical and ecosystem model. *Biogeosciences*, 12, 4447– 4481. https://doi.org/10.5194/bg-12-4447-2015

Gnanadesikan, A., Kim, G. E., & Pradal, M.-A. (2019). Impact of colored dissolved materials on the annual cycle of sea surface temperature: Potential implications for extreme ocean temperatures. *Geophysical Research Letters*, 46, 961– 869. https://doi.org/10.1029/2018GL080695

Groeskamp, S., & Iudicone, D. (2018). The effect of air-sea flux products, shortwave radiation depth penetration, and albedo on the upper ocean overturning circulation. *Geophysical Research Letters*, 45, 9087– 9097. https://doi.org/10.1029/2018GL078442

Hill, V. J. (2008). Impacts of chromophoric dissolved organic material on surface ocean heating in the Chukchi Sea. *Journal of Geophysical Research*, 113, C07024. https://doi.org/10.1029/2007JC004119

Jolliff, J. K., & Smith, T. A. (2014). Biological modulation of upper ocean physics: Simulating the biothermal feedback effect in Monterey Bay, California. *Journal of Geophysical Research: Biogeosciences*, 119, 703– 721. https://doi.org/10.1002/2013JG002522

Juhls, B., Stedmon, C. A., Morgenstern, A., Meyer, H., Hlemann, J., Heim, B., Povazhnyi, V., & Overduin, P. P. (2020). Identifying drivers of seasonality in Lena River biogeochemistry and dissolved organic matter fluxes. *Frontiers in Environmental Science*, 8, 53.

Kim, G. E., Gnanadesikan, A., Del Castillo, C. E., & Pradal, M.-A. (2018). Upper ocean cooling in a coupled climate model due to light attenuation by yellowing materials. *Geophysical Research Letters*, 45, 6134– 6140. https://doi.org/10.1029/2018GL077297

Kim, G. E., Gnanadesikan, A., & Pradal, M.-A. (2016). Increased surface ocean heating by colored detrital matter (CDM) linked to greater Northern Hemisphere ice formation in the GFDL CM2Mc ESM. *Journal of Climate*, 29(24), 9063– 9076. https://doi.org/10.1175/JCLI-D-16-0053.1

Kim, G. E., Pradal, M.-A., & Gnanadesikan, A. (2015). Quantifying the biological impact of surface ocean light attenuation by colored detrital matter in an ESM using a new optical parameterization. *Biogeosciences*, 12, 5119– 5132. https://doi.org/10.5194/bg-12-5119-2015

Pefanis, V., Losa, S. N., Losch, M., Janout, M. and Bracher, A. (2020): Amplified Arctic Surface Warming and Sea Ice Loss Due to Phytoplankton and Colored Dissolved Material, *Geophysical Research Letters*, 47, e2020GL088795. doi: 10.1029/2020GL088795

Soppa, M. A., Pefanis, V., Hellmann, S., Losa, S. N., Hölemann, J., Martynov, F., Heim, B., Janout, M. A., Dinter, T., Rozanov, V., & Bracher, A. (2019). Assessing the influence of water constituents on the radiative heating of Laptev Sea shelf waters. *Frontiers in Marine Science*, 6(221), 1– 13. https://doi.org/10.3389/fmars.2019.00221

---

## Author Comment (AC1) · 25 May 2021

First of all, we would like to thank two referees for the thorough reviews of our manuscript. We followed every suggestions and think that the manuscript has improved considerably.

In the following, we respond to the referee's specific remarks. Remarks are shown in blue and our response in black.

Section numbers used in our response refer to the revised manuscript.

**Review #1 (M. Baird):**

*Major comments.*

*1. The authors are occasionally loose with the use of CDOM vs. CDOM absorption and this becomes confusing. For example, p2 L17 is CDOM absorption; L26 'amount of CDOM'. Is this a load, concentration, rate of absorption?*

We went through the manuscript (MS) and clarification whether CDOM concentration/content or CDOM absorption is meant. We also clarified in the model section (2.1.3) that we consider CDOM as a substance in the water with different properties. One of these properties is light absorption.

*2. While it is not entirely clear, the authors appear to attribute the improvement of their CDOM equation compared to the salinity relationship to the inclusion of nonconservative behaviour CDOM. Instead, I suspect the majority of the improvement is due to the use of the 65 stations to better set the inflow concentrations of CDOM. If they run a simulation with the non-conservative terms set to zero they would be able to quantitatively compare the importance of one over the other. The comparison should also include a complete description of the salinity-CDOM absorption parameterization so that we understand the comparison. For researchers such as myself considering both options in a coastal model, this would double the value of the paper.*

We are sorry that the message is not obvious enough from the MS. Of course, the improvement could be achieved only due to high quality forcing data from EO methods. We made this statement more pronounced in the abstract. The diversity of riverine CDOM is also obvious from the new Fig. 4.

We performed an additional simulation with photobleaching switched off as suggested by referee #1. Results are shown in appendix 1. In appendix 1, we also provide the salt-CDOM absorption relation from Neumann, 2015. This relation is hyperbolic and therefore implicitly accounts for degradation.

*3. The authors use a neural network to determine the CDOM component of absorption at 440 nm at 65 sites. This is a key innovation. They then undertake a convoluted set of calculations, including choosing an arbitrary 75th percentile value, in order to turn the satellite-determined absorption into a CDOM concentration which is then multiplied by kcdom in Eq. 5 to obtain the component of vertical attenuation due to absorption CDOM. Is this complicated pathway even necessary? Absorption is inherently additive. Furthermore, the degradation rate is proportional to concentration, which is itself proportional to absorption. So, could you not simply have a model tracer CDOM absorption at 440, and applying the mixing and non-conservative terms to this tracer?*

The referee is right; the model would be simpler just by considering CDOM absorption only. However, as outlined in our response to comment #1, we want to have CDOM as a "substance", eventually subject to further biogeochemical processes in the model. We clarified it in the MS.

*4. I presume the use of the 75th percentile is about trying to determine the CDOM absorption in the freshwater end member. Given that there is a hydrodynamic model, the authors could use salinity in the hydrodynamic model to determine the unique freshwater endmember for each of the 65 sites? I know you started with the feeling that salinity vs CDOM doesn't work, but I think this is because the Baltic has 65 different freshwater end members.*

Yes, the 75th percentile is assumed to represent the freshwater portion of the river water before it is mixed with sea water. We also tested 90th percentile and then decided to proceed with $75^{th}$ percentile in this first test to demonstrate the feasibility of the method. There is certainly more that can be done to finetune and streamline the process.

The combined model/EO approach to determine the CDOM endmember for the rivers seems to us not straight forward and should be subject for a follow up study. One problem is the high spatial resolution of EO data which we cannot match with a model at the moment. Nevertheless, this high resolution in EO is essential to get values from the river mouths. A further problem will be that the model is not able to reproduce river plumes one by one. Reasons are the coarse temporal resolution of runoff available for most rivers. But also uncertainties in the meteorological forcing and nonlinear features like filaments may prevent a one by one projection of the model plume on the EO plume. The idea is brilliant, but we think a sound evaluation is needed and it seems to us it is beyond the scope of this study.

*5. One of the key findings is the variability in CDOM absorption at the 65 stations. Fig. 4 illustrates this, but much more could be shown. I suggest splitting the two panels into 2 figures, and showing the map as large as Fig. 1, but with the 65 sites with symbols collected by mean absorption at 440. For some researchers this alone would be an important result.*

Following the referees recommendation, we split Fig. 4 and replaced Fig. 4b by a new map showing EO based CDOM absorption in the rivers used (new Fig. 4 in the revised MS).

*Minor comments.*

*Title: "Radiation model" might imply a more sophisticated, directional model of light. Perhaps "Optical model" is less specific?*

We followed the recommendation and changed the title accordingly.

*P1 L17 what does divergence mean in this context?*

We use absorption now.

*P2 L27 always have a space between a quantity and its units.*

Done

*P2 2nd para. Paragraph goes from discussing non-conservative behaviour (2nd sentence), conservative behaviour (4th) to non-conservative again (5th). I understand what you are trying to say, and of course the point of the paper is in part the non-conservative behaviour. Paragraph just needs a more logical flow.*

We re-phrased the paragraph.

*P3 EO processors – does this mean software, theory?*

It is a set of algorithms, we clarified it in the MS.

*P3 L9 water leaving reflectance is a tautology.*

We modified the sentence.

*P3 L12 coastal waters of Finland*

We modified the sentence.

*P5 L15 'behaves conservatively',*

We modified the sentence.

*P5 Eq. 8. Replace '2' in the equation with a parameter, the fraction of SWR in total solar radiation.*

We introduced a new parameter r.

*What is the difference between PAR(z) and I(z)?*

The referee is right, there is no difference. We substituted I(z) by PAR(z).

*Are you sure about exponential term in Eq 8.*

Yes, we think this term accounts for the light attenuation at depth z depending on model tracer concentrations above depth z. We made it more clearly in the text.

*The K_PAR in front of the integral doesn't seem write.*

Thanks, it is a typo.

*For equations, paper "Edwards, A.M. and Augerˇ AR ˇMéthé, M., 2019. Some guidanceon using mathematical notation in ecology. Methods in Ecology and Evolution, 10(1),pp.92-99." Is helpful.*

We are thankful for this hint and considered the recommendations.

*P5 L10 K_CDOM is a parameter, not a statistical relationship.*

Here, K_CDOM(S) is meant, we correct the MS accordingly.

*P5 L10 CDOM absorption?*

Yes, it should be absorption, corrected.

*P5 L14. Isn't DON part of the DOM? In which case are the last two terms in Eq. 5 double counting?*

The last term in Eq. 5 (Eq. 4 in the revised manuscript) is terrestrial CDOM. It is part of a general DOM, but we treat it separately in the model. DON, also part of DOM, is produced in the model by phytoplankton and is a separate state variable as well. We think there is no double counting. The schematic of the biogeochemical model in the appendix may illustrate the different state variables.

*P5, 2nd last line. 'depending on sun zenith angle, which is a function of latitude and time of day"*

We thank the referee for clarifying this statement.

*P6 L8 per m3*

Corrected.

*P6 L22 Whet is "Basis"?*

We changed it into fundamentals.

*P7 Title 3.3 Model configuration?*

Most parts of this section are now in a new section "Circulation and biogeochemical model". The section "Model configuration" dose not exists anymore.

*P7 A schematic of the biogeochemical model would help here.*

A schematic is given in Appendix B.

*Fig. 3b colorbar caption should be delta a(440).*

The delta appeared much too thin in the pdf. Corrected in a revised figure version.

*P9 First paragraph – this discussion needs to be more quantitative.*

We introduced quantitative measures in this paragraph.

**Review #2 (S. Losa):**

*General comments:*

*1) An edit is required for the title (please follow recommendation of reviewer 1)*

Done, see response to review #1.

*2) In the abstract, please present more precisely the evaluation results (including comparison of "the traditional" approach). How exactly did the model performance with the new light attenuation parameterisation improve, given which particular evaluation criteria?*

We extended the abstract accordingly.

*3) Introduction should be extended more intensively by references to the state-of-the-art of the investigated problem and related studies (see my specific comments), which would show the present study in line with already existing research and would further emphasise the added value.*

We have taken this recommendation into account together with the hints from the specific comments.

*4) The manuscript could benefit from a restructuring. In particular, Part 2: I would suggest to introduce/organize a separate section: 2 Methods and data and started first (Section 2.1) with model description*
*- general (MOM-ERGOM) model description*
*- Radiation (optical) model development*
*- Implementation in ERGOM*
*followed by*
*- data description (Section 2.2) including data processor etc. to prescribe required boundary conditions;*
*- and further details on the experiment set up including forcing and initial conditions and further followed by validation/evaluation metrics (Section 2.3)*

Following the recommendations of referee #2, we restructured the sections of the manuscript. We introduced a section "Methods and data" with two subsections "Model development and description" and "CDOM boundary data". Both subsections are further structured into subsubsections.

*5) Generally, I would also recommend elaborate a bit more on the results (however I do not list specific comments with respect, except for a request on quantitative estimates of the discussed correlations).*

We are more specific now, especially for quantitative estimates.

*Specific comments:*

*P1. L13-16: It would be nice to support your statements by related references (sentencewise).*

We feel that the two sentences refer to quiet fundamental knowledge. A good summary is given by Nelsen&Siegel, 2002 which we use as a reference.

*P1. L17: Please add related references in support to the statement ("Water temperature is affected by CDOM absorption as well"). For instance: Hill, 2008; Kim et al, 2015; Kim et al., 2018; Gnanadesikan et al., 2019, Soppa et al., 2019, Pefanis et al. 2020.*

We are very grateful for the comprehensive support by referee #2 and recognized the relevant literature.

*L19: Provide related references*

We added a related reference.

*P2. L2: Even for open ocean several studies showed a better representation of the light path when explicitly accounting for light absorption by chlorophyll and CDOM (Kim et al. 2015, Kim et. 2016, Groeskamp&Iudicone, 2018, Pefanis et al., 2020). Nevertheless, I agree that for coastal ecosystem it is extremely crucial (Cahill et al. 2008; Jolliff&Smith, 2014; Juhls et al. 2020).*

We added a related reference which highlights the impact of river plumes.

*P2. L3: I would suggest "parameterisation of light (penetration)" instead of "parametrization of model"*

We changed the text accordingly.

*P2. L4: "autochthonously" instead of "autochthonous"*

We changed the text accordingly.

*P2. L14-15: the authors might want to add the following references: Dutrkiewicz et al. 2015, Pefanis et al. 2020*

We added relevant references.

*P2. L15: "In relation to the Baltic Sea, a necessary prerequisite …."*

We think that a lack of riverine CDOM data is not only the case for the Baltic. In this relation we refer to Pefanis et al. (2020) who note this problem also for Arctic rivers.

*P2. L19: A rephase is required: "... we discuss the effect of the new development (proposed model extension?) on the ... Baltic Sea ... "*

We changed the text accordingly.

*P2. L21: please consider editing of this sentence.*

We changed the text accordingly.

*P3. L7-9: An edit is required for this sentence. As an example: "It utilizes an artificial neural network (ANN) first to remove ... and then to estimate ... "*

We changed the text accordingly.

*P3. L11: "gelbstoft" - please use English term :-)*

We substituted gelbstoff by yellow substances.

Done

*P3. L12: "measurements from Finland" - please provide a related reference and/or link*

The data used in the calibration have not yet been published, but the method is similar to the one described Attila et al. (2013). We also added a reference to the use of the flow-through device (Lindfors et al. 2005).

Done.

Done.

*P5. L2: I would suggest: "as in the study by Neumann et al. "instead of "proposed by"*

We changed the text accordingly.

*P5. L17-18: please provide a supporting reference to the statement.*

A supporting reference has been added.

*P5. Equation 8: please edit the integral part of the equation.*

Due to a typo, KPAR was a factor in front of the integral which have been deleted. However, we think the integral itself is correct. It accounts for the depth dependence of the contributing concentrations in the model. We changed the manuscript and explain it in the manuscript.

*P6. L16-21: consider combining the corresponding text in one paragraph.*

We merged the paragraphs.

Done.

*P7. L19: CDOM as a product of phytoplankton is neither considered. Right?*

That's right. In the current version, we consider terrestrial CDOM only since in the Baltic Sea *in situ* produced CDOM is a small fraction of total CDOM. We made this fact more pronounced in the related paragraph.

*Part 4: since there is no a discussion part, the best title would be "Results and discussion" (not just "Results")*

The referee is right, we changed the section title.

*P7. L27-28 (second sentence of Part 4): strictly speaking it might also impact the physics, but probably not in the current set up… Somehow, it was not clear enough from Parts 2/3 if the authors consider CDOM effect on the shortwave radiation penetration (and related physical processes) in general or only as a role of CDOM absorption in attenuation of the light available for phytoplankton production/growth. Please provide required emphasises.*

We noted the small, and therefore not shown, impact on physics in section "impact on biogeochemistry (3.2)". The small effect is due to the fact that CDOM light absorption was also included in the former model version but with the salinity approximation for CDOM. In section "model configuration", we mention now that the estimated short wave absorption feeds into the ocean model as well.

*P7. L28-32: these sentences should belong to the "Method" part.*

We think it fits best here (Sec. Results and discussion), since it describes how we prepared the various observational data for comparison with model data. We simply used available observations and think that an own section on these data is not useful.

*P9. L4-3: "the correlation is low" – please provide quantitative estimates if possible.*

Done

*P9. L4-5: "correlation improves …" – provide the quantitative estimates (r = …)*

Done

*P9. L13-20: The text belongs to one (joint) paragraph.*

Done.

*P9. L31-32: might belong to the "Method"*

We think it fits best here. We simply used available observations and think that an own section on these data is not useful.

*Part 5 Conclusions: reads rather as "Summary and conclusions"*

The title is changed into "Summary and conclusions":

*P10. L7: editing is required for "an approach for light absorption" As a suggestion: "...an approach for accounting for the light absorption due to ..." Or "...an approach for approximating/considering the light absorption due to ..."*

We changed the sentence accordingly.

*P10. L10: "A common approach uses CDOM-salinity relationship for …" Readds too general, please rephrase, since not all studies in existence use CDOM-salinity relationships to represent CDOM in models.*

We agree with the referee and changed it into "an often applied …".

*P12, L5: the authors might want to refer to the study by Dutkiewicz et al. (2015).*

We are grateful for this hint and added the reference.

*Figures*

*Figures 2, 4: to improve the quality of the figure please increase the size of the font used.*

We revised the figure.

*Figure 3: increase the size of the figure panels.*

We increased the figure panels.

*Figure 6 caption: "… based on its relation to salinity" instead of "based on salt."*

We changed the caption accordingly.

---

## Author Response (AR2)

IOW, Seestraße 15, 18119 Rostock

Dr. Thomas Neumann

Senior Scientist

Seestraße 15
D-18119 Rostock
phone: +49 381 51 97 130
fax: +49 381 51 97 440
www.io-warnemuende.de
thomas.neumann@
io-warnemuende.de

Rostock, 14.07.2021

Paul Halloran
Topical Editor
GMD

**Revision gm-2020-318**

Dear Paul Halloran (GM editor)

Hereby, I submit the revised version of the manuscript gm-2020-318

**Optical model for the Baltic Sea with an explicit CDOM state variable: a case study with Model ERGOM (version 1.2)**

Thomas Neumann, Sampsa Koponen, Jenni Attila, Carsten Brockmann, Kari Kallio, Mikko Kervinen, Constant Mazeran, Dagmar Müller, Petra Philipson, Susanne Thulin, Sakari Väkevä, and Pasi Ylöstalo

First of all, we would like to thank you and the two referees for the careful consideration of the manuscript and the helpful comments. We followed your and the referees' suggestions.

We changed the sentence in the abstract as you proposed and increased the size of the panels in figures 3 and 6. Thank you for time and effort to consider the submitted manuscript.

With best regards,

Thomas Neumann

**Appendix**

In the following, we respond to the editor's and referee's specific remarks. Remarks are shown in blue and our response in black.

**Editor and review #1:**

*The reviewer highlighted, and I agree, that the sentence "We show that the light absorption by CDOM in the model can be improved considerably compared to traditional approaches where, e.g., CDOM is estimated from salinity." reads too conversationally. Please can this be revised to read something like "We show that the light absorption by CDOM in the model can be improved considerably in comparison to ..."*

The sentence in the abstract now reads: "We show that the light absorption by CDOM in the model can be improved considerably in comparison to approaches where CDOM is estimated from salinity."

**Review #2:**

*I would just suggest to increase the size of the panels of figures 3 and 6.*

We increased the size of the panels in figures 3 and 6.

[revised manuscript text omitted]